# Community Participation in and Perception of Community-Directed Treatment with Ivermectin in Kinshasa, DRC

**DOI:** 10.3390/tropicalmed4030109

**Published:** 2019-07-19

**Authors:** Jean-Claude Makenga Bof, Paul Mansiangi, Horlyne Nsangi, Éric Mafuta, Isabelle Aujoulat, Yves Coppieters

**Affiliations:** 1School of Public Health, Université Libre de Bruxelles (ULB), Route de Lennik 808, 1070 Brussels, Belgium; 2School of Public Health, Faculty of Medicine, Université de Kinshasa (UNIKIN), Avenue de l’Université, Lemba, Kinshasa, Democratic Republic of the Congo; 3Faculty of Public Health, Université Catholique de Louvain (UCL), Clos Chapelle-aux-Champs 30, 1200 Woluwe-Saint-Lambert, Brussels, Belgium

**Keywords:** Onchocerciasis, Ivermectin, communities, participation, Binza Ozone, Kinshasa

## Abstract

The success of community-directed treatment with Ivermectin (CDTI) depends on active community participation. We conducted a case study nested in a cross-sectional study in the Binza Ozone Health Zone (ZS) in Kinshasa, Democratic Republic of Congo, in order to investigate community’s knowledges and perceptions of onchocerciasis and on all CDTI’s aspects. We interviewed 106 people aged 20 and over, purposively selected, through eight individual interviews and 12 focus groups. Themes used for collecting data were drawn for the Health Belief Model and data were analyzed using a deductive thematic approach. The term onchocerciasis was unknown to participants who called it “Mbitiri”, the little black fly, in their local language. This disease is seen as curse put on the sufferer by a witch and perceived as a threat because of the “Mbitiri” bites. The afflicted participants were reluctant to seek treatment and preferred traditional practitioners or healers. CDTI is considered devastating because of adverse effects of ivermectin as well as inefficient after occurrence of deaths. This explains the low level of community adhesion and participation to this strategy. Recruitment procedures for community distributors are poorly understood and awareness and health education campaigns are either non-existent or rarely carried out. Nevertheless, the latter should be regularly done.

## 1. Introduction

Onchocerciasis, or river blindness, is a parasitic disease caused by *Onchocerca volvulus*, transmitted by Diptera females of the genus *Simulium* [1]. It causes serious medical and socio-economic problems. Medically, Onchocerciasis causes severe pruritus, nodules, dermatitis, depigmentation and, in the final stage, blindness [2,3]. On the socio-economic level, it leads to the abandonment of work, the depopulation of fertile lands, and various other consequences up to the loss of social status (rejection and stigmatization), especially when the afflicted becomes blind [4]. It leads to absenteeism among school children and deprives them of their youth since they become, in certain regions, the “seeing eye guides” of blind people [5]. Onchocerciasis is one of eleven neglected tropical diseases (NTDs), recently targeted by the World Health Organization (WHO) for elimination out of the twenty existing ones, affecting poor populations living in the tropics and subtropics areas [6]. According to the WHO, around 120 million people worldwide are at risk of Onchocerciasis, among which 96% are in Africa [7]. Out of the 38 countries with endemic Onchocerciasis, 31 are located in sub-Saharan Africa, six in North and South America, and one in Asia [8]. More than 99% of infected people live in 31 African countries. The Global Burden of Disease Study estimated in 2017 that there were 20.9 million prevalent *O. volvulus* infections worldwide: 14.6 million of the infected people had skin disease and 1.15 million had vision loss [8]. This high prevalence demonstrates that Onchocerciasis is a priority public health problem in the countries concerned, usually with serious socio-economic consequences [7,8]. In July 2016, after successfully implementing eradication activities for decades, Guatemala became the fourth country in the world declared free from Onchocerciasis, after Colombia in 2013, Ecuador in 2014, and Mexico in 2015 [8].

In sub-Saharan Africa, Onchocerciasis is the second leading cause of preventable blindness; according to data of WHO published in 2003, it causes visual impairment in nearly 500,000 people and blindness in nearly 270,000 people [9]. 

In the Democratic Republic of Congo (DRC), using the Rapid Epidemiological Mapping of Onchocerciasis (REMO) method, the map of Onchocerciasis reveals that the disease is present in the 26 provinces of the country, at different levels of endemicity [10,11]. Nearly 38 million people are at risk [11]. Nodular prevalences range from 1% to 100% [10,11]. According to the Global Burden of Disease Study 2013, DRC had the largest number of cases of Onchocerciasis, estimated at 8.3 million [12]. Moreover, it is the country most affected in terms of blindness as secondary affect to the disease [12,13]. Current efforts in DRC to eradicate Onchocerciasis revolve around the community-directed treatment with Ivermectin (CDTI) strategy, initiated by the African Program for Onchocerciasis Control (APOC) and implemented in each afflicted country by a National Program for Onchocerciasis Control (NPOC). This strategy involves communities in the fight against this disease, building trust and partnership working between health care services and communities, and strengthening national health systems [14,15]. Community participation is a process in which individuals and families take charge of their own health and well-being, as well as that of the community they belong to [14,15]. It is an indispensable element in the success of CDTI and in the development of the program to fight this disease, as reported by Hopkins [15].

The involvement of individuals and communities in the fight against Onchocerciasis has led to an improvement in therapeutic coverage, which has increased from 1968 communities in 2001 to 39,100 in 2012, from 2.7% (2001) to 74.2% (2012). Geographical coverage has increased from 4.7% (2001) to 93.9% (2012) [10,16]. The success or failure of this CDTI strategy depends on how communities become accustomed to taking ownership and to assuming responsibility in implementing this project. The WHO Technical Advisory Committee states in its report that the lack of community involvement in CDTI led to low treatment coverage, particularly in Chad and Nigeria [17]. On the one hand, there are several districts in the city of Kinshasa endemic with Onchocerciasis which have benefited from the CDTI program for several decades and, on the other hand, 3 Onchocerciasis hotbeds: Mont-Ngafula, Nsele, and Kinsuka pêcheurs. However, the results of the studies carried out in the health zones in urban areas showed a weak involvement of community volunteers in primary health care activities. Also, the involvement of the population in terms of community participation was low [18].

After 15 years of CDTI in DRC, there is little information on community adhesion and participation in CDTI and questions remain both in terms of disease’s knowledge or perception and of CDTI. The purpose of this study was to investigate the community’s knowledges and perceptions of onchocerciasis and on all CDTI’s aspects after several years of CDTI. The results of this study could lead to the development and dissemination of more appropriate awareness and health education messages.

## 2. Methodology

### 2.1. Type of Study

This is a case-study nested in a cross-sectional study carried out in households in the Health Zone (HZ) of Binza Ozone in Kinshasa which is the capital of the Democratic Republic of Congo (DRC) [19]. The HZ in DRC is equivalent of Health District elsewhere. In DRC, HZ is divided into health areas (HA), supplied in health care by a health center. The study involves the population of the urban health areas (HA) of Kinsuka pêcheurs, Lukunga, Manenga, and Mangana, living along the left banks of Congo River, in the beginning of rapids of Kinsuka. These HA were targeted because they have been benefiting from Onchocerciasis training activities and the implementation of CDTI project activities since 2003. 

According to WHO, HZ is defined as the operational unit whose development remains the prerequisite for implementing health policy, HA is a well delimited geographic entity composed of several villages in rural areas/several streets in urban environments, established according to sociodemographic affinities, and the district is the level where health policies and health sector reforms are interpreted and implemented [5]. 

### 2.2. Study Environment

The HZ’s Binza Ozone is located in the municipality of Ngaliema and is crossed by the rivers Lukunga and Binza, joining the Congo River in its non-navigable part of Kinshasa. Binza Ozone is considered as a meso-endemic HZ. A meso-endemic zone is an area in which prevalence of nodules is between 20%–39.9% and presents a danger for the population due to the proliferation of small black flies. Those flies are the main cause of river blindness and constitutes the urban hotbed of Kinsuka pêcheurs [18]. In other words, the Binza Ozone HZ is part of Kinsuka Pêcheurs urban hotbed and includes 113 communities that benefited from CDTI. In 2011, this hotbed had a therapeutic coverage of 71.6% in CDTI. In 2013, health-wise, malaria was the most common disease (30%). Onchocerciasis and other filariasis accounted for about 0.1% and mainly affected the working population, namely those over the age of 15 years (65% of filariasis cases) [20]. In this hotbed, a study showed low transmission of Onchocerciasis and a high biting rate in 2012 i.e., the biting rate were about 12 times higher than that observed 27 years ago, in the same focus [18]. 

### 2.3. Population Studied, Samples

The study focused on community members and key informants. According to the WHO, the community is defined as a group of people who often live in a well-defined geographical area, share a culture, values and norms, and who have a place in a social structure that is consistent with relationships that the community has created over a period of time. [21]. A key informant or interviewee is an individual chosen from a social system that reflects the thinking of a group. These are people who have in-depth knowledge of the problem and the target audience. These people provide an overall picture of public opinion, practices and attitudes. In our study, they were chosen from among the community volunteers [22]. A Community volunteer is a volunteer person, male or female, living in the village or street, chosen by the inhabitants of this entity, who bridges the gap between individuals, members of a family, and the health service [23].

It studied members of the community aged at least 20 and over, whether they were the head of household or not, having lived at least 5 years in the areas of the HZ’s Binza Ozone and having participated in the CDTI activities. Community members were invited to participate in focus groups. In respect of key informants, this included anyone with some administrative or public policy responsibility at the HA level in the HZ’s Binza Ozone. 

Participants were selected using a purposive sampling strategy. The research team first identified the HA that benefited from APOC activities. Then, after collecting information from the community volunteers and members of the Development Committee in each HA, the team listed the households with eligible persons and finally, it proceeded to the recruitment of the participants in the focus groups, taking into account age and sex to form homogeneous groups. Therefore, four different categories of participants were selected to support a maximum of variation strategy: Men >30 years, women >30 years, young women aged 20–29 years old and young men aged 20–29 years old. These people were contacted by community volunteers and once the study was explained to them, they were invited to participate. The focus groups size has been set at a minimum of 8 to 10 people for a maximum of 4 focus groups in each HA.

The sampling procedure described above was also followed for key informants who are community leaders (community volunteers, musicians, pastors, neighborhood leaders, department heads, etc.). A list of these key informants has been compiled by HA in collaboration with community volunteers and members of the HA Development Committee. For the in-depth interviews, we wanted to recruit at least two key informants per HA in order to reach eight. The research team presumed that these data were optimal and after reaching 12 focus groups and eight key informants, data collection would not have produced any new information and no new information would have emerged.

### 2.4. Collection of Data

Data were collected from 01/10/2016 to 01/01/2017 from focus groups based on household participants and in-depth individual semi-structured interviews for key informants. These different data collection techniques were used concomitantly. The focus groups aimed to explore the community’s knowledges and perceptions of onchocerciasis, CDTI and its participation in control activities. In this context, group dynamic discussion regarding onchocerciasis and CDTI was sought between the participants and furthermore, an educational objective was pursued. In this way, the participants were led during the discussions to reflect on their potential role as actors in the fighting against the disease in their community as well as on the knowledge, attitudes, and practices that they had to have to properly assume their participation. In-depth interviews were conducted with categories of people for whom there were few respondents and for whom a group discussion was difficult to organize, such as politico-administrative authorities and some community leaders. Such interviews also sought to obtain an individual’s opinion but also the overall vision of the community in terms of opinion leaders and other key informants. Focus groups and interviews were conducted, respectively, according to an appropriate guide.

These techniques have been favored because they allow, on the one hand, rapid accumulation of information and, on the other hand, to capture the meaning and perceptions of the groups’ behaviors or social practices. They facilitate the survey of opinions, attitudes, motivations, and behavioral factors related to community adhesion on CDTI.

Participants in both individual interviews and focus groups, in which participants, separated by sex (men and women) and age (young and adults), had to answer questions based on the interview guide. The interview guide was pre-tested for adjustments and was translated into the local language (Lingala) to allow each respondent to freely choose the language (French or local) in which he or she could express his or her ideas the best way. Focus groups were conducted in community settings such as schools, churches, and quiet places chosen by community members. The average duration of the focus groups was one hour to one-and-a-half hours. Key informant interviews were conducted in a location of their choice. The focus group recordings and interviews were transcribed verbatim from the local language (Lingala) into French, respecting idioms and field notes taken by the members of the research team. These interviews lasted an average of 45 minutes to an hour.

### 2.5. Themes

Themes were adapted from Health Belief Model (HBM) [24,25,26]. The HBM is the first model built with the objective of explaining what motivates an individual to adopt a healthy behavior. [24,25,26]. We chose HBM because it was developed to understand why people did not undergo tests to diagnose diseases early (such as screening tests) but also to explain and predict whether individuals adopt preventive behaviors (e.g., condom use) [24,25,26]. This model is based on the premise that health is an important value for the person and that the decision to behave is rational. [24,25,26]. It is premised on the hypothesis that an individual, in order to initiate or modify health behavior, must first: perceive a disease threat and believe in the effectiveness of the health behavior to be undertaken to reduce that threat [24,25,26]. 

The questions asked during this study were in relations to the perception of disease, its clinical manifestations and of whether the disease is considered as a public health problem or not; to the knowledge of the disease vector in the community and the disease’s consequences; to the perception of aspects of CDTI, awareness campaigns and health education; to the perception on CDTI monitoring, evaluation, planning, recruitment of community distributors and organization of CDTI, as well as the wishes expressed by participants.

### 2.6. Data Processing

The data were analyzed according to the deductive approach of thematic analysis. This approach consisted firstly, after the data collection, in transcribing all the recordings of focus groups and interviews while respecting the Lingala verbatim in French. Then the transcription was made by the people who conducted or led the focus groups or interviews. To become familiar with their content, finally these transcripts were read and re-read by two members of the research team. At the end of this familiarization, each of them proposed a codification guide mainly based on themes drawn on the Health Belief Model, which was consensually integrated into a single guide after debate. Based on this codification guide, the transcripts were imported and organized using Atlas-ti 8.0 software (ATLAS-ti GmbH, Berlin). The results produced by this software have been synthesized to generate following each theme. During this process, when a new theme emerged, it was integrated into the codification guide. 

## 3. Ethical Considerations

The study was approved by the Ethics Committee of the School of Public Health of the University of Kinshasa (Approval Number: ESP/CE/090/2016). Each study participant was informed of the purpose, procedures, risks and benefits of the study and provided informed consent before participating. The majority of participants gave their consent verbally which is typical behavior of the local community who are reluctant to sign documents. The confidentiality of the data collected and the anonymity in the processing, analysis, and presentation of data have also been respected. The study respected the principles laid down in the Helsinki II Declaration. 

## 4. Results

A total of 106 people were involved in this study, including 8 who participated in individual interviews and 98 participated in focus groups (Table 1 and Table 2). The following results represent a summary of the opinions expressed by the participants and are illustrated by quotes from the transcript of both individual and group interviews. 

### 4.1. Perception of Disease, Its Clinical Manifestations, and of Disease As A Public Health Problem

Onchocerciasis is perceived by participants as a curse cast by a witch. Participants do not know it in terms of illness. Instead, they speak of a small black fly “Mbitiri”, responsible for the transmission of Onchocerciasis. The “Mbitiri” disease is perceived as a threat because of the black fly bites. Most sufferers are reluctant to seek treatment at health centers (HC) and prefer to go see the traditional practitioners or healers. 

Participants are not specific about the origin of the disease (environmental factors or witchcraft). Participants illustrated this perception as follows: *“… when a witch casts a spell on you, you can catch several diseases, including the “Mbitiri’s Disease”. The witches are there to make people suffer, that is why they can send the “Mbitiri” to sting you to hurt you. This is the reason why harm due to “Mbitiri” bites can only be properly handled by a healer or traditional practitioner …”* (Focus group woman/health area of Lukunga). However, they do not have correct information regarding clinical signs of the “Mbitiri” disease. When they are mentioned, they are similar to those of Onchocerciasis, but are perceived by the participants as being diseases in their own right. Namely: pruritus (itching), skin lesions, swelling, conjunctivitis, wounds caused by scratching, breast swelling in girls, jaundice of the eyes in children, fever and chores body. In contrast, blindness is rarely mentioned by participants. Few believe that these diseases are more common among fishermen and farmers exposed to “Mbitiri” bites. The following is a statement of participants which illustrate this observation: “… *here we call this disease: Mbitiri disease because we do not know the scientific name of the insect, it often appears among fishermen and farmers …”* (Focus group woman/health area of Kinsuka pêcheurs).

Onchocerciasis is not perceived as a health problem but rather as a nuisance due to black fly bites worrying the community. This perception is illustrated by the following statement: “… *we ask the State to clean up the environment (channel the water and weeding) and make creams available to us that can be applied to the skin to prevent the bite of blackflies that concerns us gravely …*“ (Focus group man/health area of Kinsuka pêcheurs).

Young girls are not only concerned with the nuisance caused by the bites of “Mbitiri” but also the stains left on the body following wounds due to scratch marks. The following remarks illustrate this observation: *“… these blackfly bites leave black spots on the skin, diminish our beauty and make the girl less likely to be loved by a boy …”* (Focus group girl/health area of Manenga).

Within this community, the disease with its term “Onchocerciasis” or the “Mbitiri” disease are not subject of discussion. For participants, the biggest health problem in the community remains malaria followed by other diseases such as cough and diarrhea. The following remarks clearly illustrate this observation: *“… here the members of our community suffer only from malaria, sometimes cough, and more often from diarrhea. These are the diseases that are the focus of our daily concern. We rarely discuss the “Mbitiri” disease among ourselves it is only the bites of this fly that worry us the most. These bites cause an indescribable nuisance for those affected …”* (In-depth interview/health area of Manenga).

The participants did not discuss the consequences of Onchocerciasis in order to give their perceptions. Yet, a small number of older (residents of the area for several decades) and key informants are suffering from Onchocerciasis, becoming blind over time. 

### 4.2. Knowledge of the Disease Vector in the Community and the Disease’s Consequences

The Mbitiri is described by participants as a small black fly, short in size and different from mosquitoes. In addition, they specify that this little fly bites in the late morning and the evening, causing an extreme nuisance, which is the main concern of the inhabitants of the urban hotbeds of Kinsuka. Some of them are of the opinion that, apart from the “Mbitiri” disease, black flies also transmit malaria, sleeping sickness and typhoid fever. After analyzing their remarks, it is clear that the participants do not make any connection between the bites of black flies and Onchocerciasis. The following remarks clearly illustrate this observation: *“… the Mbitiri usually bites in the morning and in the evening. After scratching due to the nuisance of the bites, I see that malaria starts, sometimes sleeping sickness and sometimes typhoid fever …”* (In-depth interview/health area of Manenga).

Focus groups participants do not recognize the existence of the disease in this HZ. The word “Onchocerciasis” is initially difficult to pronounce and almost unknown. In this urban hotbed of Kinsuka, there is only the “Mbitiri” disease that is mentioned. During the study, we observed other manifestations of Onchocerciasis, including leopard skin in a fisherman who requested Ivermectin and filarial skin lesions in a mother. Onchocerciasis does exist in this HZ but is known by its other name “Mbitiri” but whose clinical signs are not known to the population. This is illustrated by the following statement: *“… the illness you’re talking about does not exist here. Here we only have the “Mbitiri’s disease”. This disease is not really dangerous, it’s like malaria, typhoid fever, cough, and many others …”* (Focus group girl/health area of Mangana).

Although the main trends in this theme are the lack of knowledge of the disease and its consequences, some participants in the focus group expressed their views as follows: the need to inquire with HC to get an idea about the number of cases Onchocerciasis in the community; additional information or explanations from health professionals who only talk about microbes in the blood instead of Onchocerciasis. Some participants deny the very existence of the disease in their community. This is illustrated by the following statement: *“… this disease does not exist in our neighborhood and we do not see people who suffer from it …”* (focus group young boy/health area of Kinsuka pêcheurs).

Thus, despite knowing the vector that transmits the “Mbitiri disease” in the community, we encounter several people in this urban hotbed of Kinsuka who have incorrect information on Onchocerciasis. Regarding the consequences of Onchocerciasis, it appears from the group discussions that participants do not know about them. This is illustrated by the following statement: *“… when we talk about Onchocerciasis, it’s hard to answer questions because the majority, especially young people, do not know about the disease …”* (Focus group young boy/health area of Manenga).

Only a minority of older participants who have lived in the area for several decades and key informants know that Onchocerciasis sufferers may become blind in the long run.

### 4.3. Perception of Aspects of CDTI, Awareness Campaigns, and Health Education

Coordination and Services of NOCP’s CDTI strategy to eradicate Onchocerciasis has set up an annual treatment in each zone where Onchocerciasis has been declared endemic. However, there is not a great uptake in Binza Ozone HZ to motivate community members to take Ivermectin. In fact, participants perceive taking Ivermectin as a remedy for other ailments and refuse to take it because of side effects. They expressed this as follows: *“… when we take Ivermectin, very often eyes, ears and the whole body start to swell, there is increased pruritus, loss of consciousness, abortion happen, vomiting, and many other side effects …; that’s why we refuse to take it …”* (Focus group girl/health area of Lukunga). Added to this are rumors about the product namely: *“… a given drug without a laboratory test beforehand, after only taking the weight and measuring the size of the person; a free drug producing adverse effects; a medicine distributed by poorly trained community volunteers; a poison because the media does not talk about it; finally, it will lead to other diseases, like AIDS that appeared after taking a vaccine …”* (Focus group girl/health area of Kinsuka pêheurs).

Some community members of the hotbeds of Kinsuka wondered why some immunization campaigns were highly publicized, while others, such as CDTI, were not. This is illustrated by the following statement: *“… during the polio campaign and mosquito nets, we saw banners everywhere, messages were always on TV and radio, but for CDTI there are no message in the media, not even a banner in the neighbourhood …”* (Focus group man/health area of Manenga). 

Respondents perceive the CDTI strategy as a barrier to eradication of the “Mbitiri” disease because of the active non-involvement of community members avoiding the side effects of Ivermectin. CDTI is considered devastating because of adverse effects and ineffective in some cases after deaths occurred. Hence the low participation of the population in CDTI.

A key informant said: *“… In the first distribution campaign, Ivermectin caused side effects to such an extent that the population refused the product. These side effects have hindered the integration of the CDTI strategy in the community. Since then, it has remained the business of community volunteers and the Health Zone …”* (In-depth interview/health area of Lukunga).

In this area, this strategy is literally called “Mangwele ya mikolo”, which means adult vaccine: *“… Ivermectin is a vaccine for adults because it is administered once a year and distribution is door-to-door by measuring the weight and height of the person …”* (Focus group woman/health area of Manenga). 

Interviewees are informed about the CDTI project introduced in the Kinsuka district. The known element is that some people visit households to give drugs against the “Mbitiri” disease. Most do not have correct information in respect of awareness and even less about health education. 

Here are the stories of some participants in respect of the CDTI project: *“… When a group of Americans passed through the Kinsuka neighborhood they noted that fishermen and farmers were scratching a little too much. They then proceeded to take samples and skin biopsies. A few months later, they took the results to the central office, which confirmed the presence of the “Mbitiri” disease. They informed and involved the Ministry of Health and then the Central Office of the Binza Ozone Health Zone located at the rive hospital, bringing them a large batch of Ivermectin. The Central Office in turn chose the districts that will benefit from mass treatment with Ivermectin. This is how our health center Galilee was selected for being a recipient of Ivermectin. The health center in turn recruited and trained Ivermectin distributors; then the latter carried out a field trip to take account of the numbers and to raise awareness among the population about Onchocerciasis and Ivermectin. It is after all this that the product has been given to us, which should be taken in the presence of these distributors …”* (Focus group man/health area of Mangana).

### 4.4. Perception on CDTI Monitoring, Evaluation, Planning, Recruitment of Community Distributors and Organization of CDTI As Well As the Wishes Expressed by Participants

Recruitment procedures for community distributors are unknown to participants. The participants say that this program depends on the HZ and HC and that the population is just submitting to what the health structures decide. It also appears that community distributors of Ivermectin are not properly trained and are chosen from community volunteers; most of them quickly gave up because of low motivation. They expressed this in these terms: *“… After the results from the survey in the hotbeds of Kinsuka, we were invited to the Central Office of the Health Zone (COHZ) for training in Onchocerciasis and Ivermectin treatment. The COHZ asked us to recruit community distributors. At the health area level, we asked the president of the Development Committee to recruit these community distributors from among community volunteers. Then there was a briefing of these community distributors …”* (In-depth interview/health area of Mangana).

Study participants report that they have no authority to monitor and evaluate CDTI. They say they have no room for maneuver with no claim to make on what is already decided at the highest level. An informant stated this as follows: *“… Who are we to monitor and evaluate CDTI? We have no power to do it. Everything is already planned in advance and all this is imposed and dictated. Once their mission has been carried out, all those who work on behalf of the NOCP or CDTI, pack up their belongings and disappear …”* (In-depth interview/health area of Lukunga).

It appears that people do not have correct information regarding the procedure for setting the period and method of distribution of Ivermectin. A key informant states this thusly: *“… It is the central office that sets the mode and date of distribution taking into account the timing of the Ivermectin supply by the partners …”* (In-depth interview/health area of Kinsuka pêcheurs).

For the participants, a large-scale awareness campaign reassures them about the quality and the source of the medicines distributed. The nuisance being the greatest concern, a vector control would be a relief to this community of the urban hotbeds of Kinsuka.

The following testimonials illustrate this: *“… An awareness campaign broadcast by the media (radio and TV) just like for other vaccination campaigns …”* (In-depth interview/health area of Kinsuka pêcheurs); *“… The return to the insecticide spray system and the clean-up of the environment can solve the problem …”* (In-depth interview/health area of Mangana).

## 5. Discussion

The present study aimed to investigate the community’s knowledges and perceptions of onchocerciasis and on all CDTI’s aspects after several years of CDTI in the Binza Ozone HZ. On the one hand, for most respondents, the perceptions of the disease called “Onchocerciasis” were incorrect and, on the other hand, community participation in CDTI was found to be low. Indeed, due to lack of information, the population avoids the side effects of Ivermectin. 

### 5.1. Generalizations of the Diseases

The term “Onchocerciasis” does not mean anything in this community. Participants who do not have correct information about this disease, also have a wrong perception of their way of life and culture. These results are similar to those found by Weldegebreal F et al. (2016) in Quara District, Ethiopia [27], by Adeoye AO et al. (2010) in the southwest of Nigeria [28] and Dimomfu BL et al. (2007) in the Central African Republic [13]. These authors have demonstrated in their studies that correct information about Onchocerciasis is beyond the reach of participants [13,27,28]. Respondents believe that a person affected by the disease is the victim of a curse or evil power caused by an act such as theft, adultery, or any other behavior that violates the good morals of the community. In other words, they attributed the disease to the curse cast by witches. This result is similar to that observed by Makenga et al. (2017) which indicated that external factors such as evil spirits, witches or fetishes were responsible for the illness of a member of their family [5]. As a result, the community usually uses a traditional practitioner or healer to treat the disease. This result corroborates that of Adeoye et al. (2010) showing that in Nigeria, belief in magic and witchcraft is widespread [28]. These authors asserted that traditional healers in Nigeria were a cultural force of great importance, to the point of suggesting that the government recognize them for their practice. [28] Ivermectin is not perceived as a drug curing Onchocerciasis by the participants but on the contrary, as a remedy causing other ills to the community of the urban hotbeds of Kinsuka. Moreover, because the media do not talk about it, the respondents perceive it more like a poison. This observation differs from that reported by Weldegebreal F et al. (2016) in Ethiopia reporting that according to the population, free drugs were simply useless to health [27]. 

### 5.2. Perception of the Disease

The disease is perceived as a threat because of the bites of “Mbitiri”. The symptoms of Onchocerciasis, which are seen as diseases in their own right, are not known to the participants. This perception does not seem to depart from what was observed by Lakwo TL and Gasarasi DB (2006) in Ndubi village and Mweya CN in Tukuyu, southwestern Tanzania, where respondents also did not correctly associate the bite of black fly with Onchocerciasis. Indeed, more than half of the respondents (64.1%) cannot make the link between Onchocerciasis and the bite of black fly [29]. These observations differ from those found by Weldegebreal F et al. (2016) in Quara, Ethiopia, where 57% of the respondents knew the origin of the disease, particularly through the bite of a black fly in this highly endemic study area [27]. This is not the case in our study area where Mansiangi P et al. (2012) emphasized the low endemicity [18]. 

Respondents do not consider Onchocerciasis a major public health problem. This is contrary to the finding established in 2012 by Manianga C in Matamba, Kasaï-Oriental, DRC, where Onchocerciasis was perceived as a real public health problem in this HA [30]. The people surveyed felt that the bites of “Mbitiri” blackflies were extremely harmful. This result coincides with that of Nana Tonem H (2008), who showed that mosquito bites are, for 70.1% of respondents in her study, a great nuisance which is accompanied by itching [31]. 

CDTI was considered devastating because of the adverse effects of taking ivermectin and was ineffective in some cases following the occurrence of deaths. This result is similar to that of Weldegebreal F et al. (2016) in Quara, Ethiopia, where many of the community distributors had misconceptions about the illness and CDTI (cause, transmission, and prevention) [27]. The authors recommended that CDTI can be supported on the one hand by adequate and continuous training and on the other by education on the different aspects of the disease [27]. 

An experimental study showed that the dominant clinical symptoms of adverse reactions and toxicity of ivermectin in animals which often lead to mortality are tremors, ataxia, central nervous system depression, and coma. This study does not show the cause of its side effects [32]. Loiasis is a major obstacle to ivermectin treatment for onchocerciasis control and lymphatic filariasis elimination in central Africa. In communities with a high level of loiasis endemicity, there is a significant risk of severe adverse reactions to ivermectin treatment. Information on the geographic distribution of loiasis in Africa is urgently needed but available information is limited as confirmed by Zouré et al. (2011) [33]. The real causes of these serious adverse reactions are poorly documented in the literature. However, according to existing studies, co-endemicity onchocerciasis loiasis is the main and most frequent cause of serious adverse events following ivermectin administration. Kuesel confirmed that the severe adverse reactions to ivermectin in people highly infected with Loa loa also delay progress towards elimination of lymphatic filariasis (LF) in Africa where the ’standard of control’ is co-administration of ivermectin and albendazole. People with LF do not directly benefit from ivermectin treatment since the LF’s symptoms are due to the macrofilariae. Consequently, use of ivermectin in areas co-endemic for LF and Loa loa, but not for least mesoendemic for onchocerciasis, does not have an acceptable risk-benefit ratio. The provisional strategy for LF control in Loa loa co-endemic areas is twice yearly treatment with 400 mg albendazole complemented with vector control (World Health Organization, 2012). This strategy is now supported by the results of a community trial of semiannual MDA with 400 mg albendazole. The study showed that the middle rate geometric microfilariae level 12 months after the first treatment was reduced by 60% relative to the pre-treatment level. Co-infections with other filariae do not appear to significantly affect the adverse reaction profile of ivermectin, probably due to the fact that parasite numbers are lower than those in heavy Loa loa infections [34].

### 5.3. Community Participation in CDTI Activities, Planning, Monitoring, and Evaluation

Active participation in CDTI in the urban hotbeds of Kinsuka is weak; the population avoids the undesirable effects of Ivermectin. Contrary results were observed by Makenga et al. (2017) in Moanza, Bandundu, DRC, where the people surveyed felt that CDTI was very helpful in eradicating Onchocerciasis [5]. 

However, in another study, similar results were observed by Makenga et al. (2018) who claimed that the adverse effects of Ivermectin were one of the causes of low population participation in CDTI, with communities refusing to take it [35]. Our observation is also similar to the results found by the NOCP in the participatory monitoring report which established low involvement of communities in CDTI activities in 2009 in the Beni-Butembo and Lubutu CDTI project as well as in the 2010 Masisi Walikale projects, Rutshuru Goma, and Ituri North [36].

Our study shows that community volunteers dealing with the distribution of Ivermectin, are insufficiently motivated by the community and even less by the state to perform their tasks properly. Similar results were obtained by Eloko G (2012), showing that the non-participation of community volunteers was linked to the low motivation of the latter and the voluntary nature of the function [37]. Our results also suggest that community distributors of Ivermectin are not sufficiently trained on the disease and CDTI strategy. Because of this lack of in-depth training, community distributors are not able to convince communities to participate in CDTI activities. A similar result has been reported by Brieger W.R., et al. (2012) which confirmed that health education was an important factor in strengthening the application of Ivermectin treatment [38]. Indeed, health education is essential to ensure satisfactory adhesion to CDTI by the population. A similar observation was made by Mukendi et al. (2019), who found that people with epilepsy in Aketi (DRC) are very willing to take ivermectin because they believe it alleviates their seizures [39], contrary to observations from Cameroon, where CDTI coverage among persons with epilepsy is very low because they are educated that epilepsy is a contraindication to Ivermectin [40]. 

Regarding the organization of CDTI in the hotbeds of Kinsuka, our results show a decrease of half of community distributors (CDs), a lack of financial support for CDs, a lack of information on the disease and the drug within the reach of communities, low involvement of CDs and the community. Contrary results have been found by York et al. (2015) in their study of factors affecting community participation in the CDTI program in Morogoro, Tanzania [41]. Indeed, these authors report that awareness of Onchocerciasis is at the highest level, methods of prevention and CDTI is high, and the population’s understanding of the taking of Ivermectin is assured. As a result, it actively is involved in CDTI activities. The same authors mention various obstacles to community participation, including a lack of understanding of the disease, fears of taking medication, a lack of confidence in the method chosen, a lack of health education materials, insufficient communication between community-based distributors and a lack of rigid drug delivery mechanisms [41].

Due to the lack of awareness campaigns and even less health education, young people do not know that onchocerciasis can cause blindness, whereas old people have witnessed it in the past. Indeed prior to CDTI, infection with O. volvulus was intense and chronic and usually progressed to blindness. However, it has been shown that use of ivermectin has drastically reduced the disease burden of onchocerciasis by averting hundreds of thousands of potential cases of blindness in treated populations. As a result, young people and recently born people cannot know how frequent blindness was in the past. Hence awareness and education campaigns are essential [8].

## 6. Conclusions

Our study carried out in the urban hotbeds of Kinsuka in the Binza Ozone HZ enabled us to reveal, on the one hand, low community participation in CDTI’s activities and on the other hand erroneous community perceptions of both the disease and CDTI. 

Indeed, the population perceives the nuisance caused by blackfly bites rather than Onchocerciasis itself as a priority health problem. Regularly awareness campaigns must be organized and focused on health education and to the CDTI adhesion in order to improve both the perception of the disease and the active participation in CDTI activities.

## Figures and Tables

**Table 1 tropicalmed-04-00109-t001:** Distribution of sample size by health area and by target for the focus group.

Targets	HEALTH AREAS (HA)				Total
	HA Lukunga	HA Manenga	HA Kinsuka Pêcheurs	HA Mangana	
Men >30 years and over	-	8	10	8	26
Women >30 years and over	8	8	8	-	24
Young women 20–29 years	8	8	8	8	32
Young men 20–29 years	-	8	8	-	16
Total	16	32	34	16	98

**Table 2 tropicalmed-04-00109-t002:** Distribution of sample size by health area and by target for in-depth interview.

Targets	HEALTH AREAS (HA)				Total
	HA Lukunga	HA Manenga	HA Kinsuka Pêcheurs	HA Mangana	
President of the CV Committee	1	1	1	1	4
Nurse	1	1	1	1	4
Total	2	2	2	2	8

CV: Community volunteers.

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
