# Peer review of "Community Participation in and Perception of Community-Directed Treatment with Ivermectin in Kinshasa, DRC"

_tropicalmed, 2019, doi:10.3390/tropicalmed4030109_

Round 1
Reviewer 1 Report
General Comments:
This is an interesting review of perceptions of CDTI in Kinshasa. The numbers interviewed however seem rather small. There are semi-quantitative analyses that could be used if the sample is big enough and this would add to the strength of the article, which is otherwise a series of quotes.
The selection of participants does seem to be rather biased towards “self-selection” but this could be a false impression.
Whereas the English needs correction in several planes “relais communautaires” does not translate to community relays which is meaningless in English. Community volunteers would be a better term.
Likewise, hotbeds would be better translated as hotspots in epidemiological terms.
You should also emphasize that this is an urban area. Problems in urban areas are often not the same as those in rural areas.
It seems the major problem from the population point of view is the biting nuisance. Blindness is not a major problem.
Specific Comments.
Line 34. You should use more appropriate references. Reference 2 is not a reference to the blindness caused by onchocerciasis
Line 38. You should get up to date with the WHO list on NTDs.
Reference 7 has nothing to do with the WHO statistics which are the one you should quote, in line 41, 44, and 45. Best to use APOC data.
Line 50, again you should use WHO data
Line 87. Explain that a Health Zone is the equivalent of a Health District in WHO terms.
Line 93 explain “meso-endemic” In fact these terms are now disappearing with the push to elimination.
Line 101. What do you mean by aggression. Is this the severe biting nuisance? Aggression is not the correct word here.
Line 117 and line 122 It is not APOC activities but CDTI
Line 134. What do you mean by theoretical saturation?
Line 170: Explain your data analysis in more detail.
Line 198: It would be good to know what proportion of the population perceive that onchocerciasis is related to sorcery and whether levels of economic development or education have any impact of perceptions. The Binza Ozone region also has some affluent areas.
Line 423 424 You have used the term Community Distributors CDs which is fine. Correct CDs in the following line
Author Response
Author's Reply to the Review Report (Reviewer 1)
Comments and Suggestions for Authors
General Comments:
This is an interesting review of perceptions of CDTI in Kinshasa. The numbers interviewed however seem rather small. There are semi-quantitative analyses that could be used if the sample is big enough and this would add to the strength of the article, which is otherwise a series of quotes.
Answer: Thank you for the suggestion. Most qualitative research theorists do not accept semi-quantitative analysis, when using focus groups as data collection strategy. Even if illustration is selected from one specific participant, the synthesis for each theme is done for all the group. We organized 12 focus groups with 98 participants (>30), gaining a huge amount of research materials.
The selection of participants does seem to be rather biased towards “self-selection” but this could be a false impression.
Answer: We do not understand self-selection. Participants in focus groups and in-depth interviews were recruited using a purposive sampling strategy. They were invited to participate in and they were free to respond or not to the invitation. We included clarification about age and sex group categories (see lines 137-140). Purposive sampling strategy means that research team used some characteristics for sampling participants (to support homogeneity of focus groups for example). This is in accordance with qualitative research methods that do not use random sampling strategy.
In qualitative, it is not the representativeness of the sample that matters but the richness and the variability of data.
Whereas the English needs correction in several planes “relais Communautaire” does not translate to community relays which is meaningless in English. Community volunteers would be a better term.
Answer: We corrected in the text by” community volunteers” as suggested.
Likewise, hotbeds would be better translated as hotspots in epidemiological terms.
Answer: we replaced hotbeds by “hotspots” in the text as suggested.
You should also emphasize that this is an urban area. Problems in urban areas are often not the same as those in rural areas.
Answer: You’re right. We insert information according to your comments as follows: “This is a cross-sectional study using qualitative methods carried out in households in the Health Zone (HZ) of Binza Ozone in Kinshasa which is the capital of the Democratic Republic of the Congo (DRC). The HZ involves the population of the urban health areas (HA) of Kinsuka pêcheurs, Lukunga, Manenga and Mangana, living along the left banks of Congo River, in the beginning of rapids of Kinsuka”. (see lines 92-95).
It seems the major problem from the population point of view is the biting nuisance. Blindness is not a major problem.
Answer: Indeed, blindness is becoming almost non-existent in urban areas of Kinshasa thanks to the distribution of ivermectin. However, blindness remains a health problem at sites such as Inga, in rural areas due to the neglect of chemoprophylaxis by this population.
Specific Comments.
Line 34. You should use more appropriate references. Reference 2 is not a reference to the blindness caused by onchocerciasis
Answer: we added the appropriate reference (number 2) related to the blindness caused by onchocerciasis.
Line 38. You should get up to date with the WHO list on NTDs.
Answer: we correct the sentence as follows: “Onchocerciasis is one of eleven neglected tropical diseases (NTDs), recently targeted for elimination by the World Health Organization (WHO) out of the twenty existing ones, affecting poor populations living in the tropics and subtropics”.
Reference 7 has nothing to do with the WHO statistics which are the one you should quote, in line 41, 44, and 45. Best to use APOC data.
Answer : We’ve adjusted reference 7 to the right line as suggested and using APOC data.
Line 50, again you should use WHO data
Answer : We confirm that these are WHO data found in the literature and published in 2003.
Line 87. Explain that a Health Zone is the equivalent of a Health District in WHO terms.
Answer: We explained as suggested : « According to WHO: HZ is defined as the operational unit whose development remains the prerequisite for implementing health policy, HA is a well delimited geographic entity composed of several villages in rural areas/several streets in urban environments, established according to sociodemographic affinities and the District is the level where health policies and health sector reforms are interpreted and implemented ».
Line 93 explain “meso-endemic” In fact these terms are now disappearing with the push to elimination.
Answer: Levels of onchocerciasis endemicity were defined as follows: sporadic zone (prevalence of nodules<10%), hypoendemic zone (10–19.9%), mesoendemic zone (20–39.9%), and hyperendemic zone (≥40%). If more than 20% of adults had nodules, mass treatment was necessary, and this figure was extrapolated to the zone as a whole. In communities where the level of nodules was less than 20%, treatment was administered via clinics
Before the transition from control to elimination of onchocerciasis, the DRC included only in treatment projects (hotbed) with a nodular prevalence of 20% or more. All Health Zones (HZ) with a nodular prevalence of less than 20% were considered hypoendemic and, therefore, not eligible for treatment. With the push to elimination onchocerciasis, the WHO currently recommends that mapping should be redone in areas of unknown status (i.e., hypoendemic areas at the time) and in non-endemic areas. Therefore, all endemic areas must be treated.
Line 101. What do you mean by aggression. Is this the severe biting nuisance? Aggression is not the correct word here.
Answer: You’re right. Aggression is not the correct translation in English of « agressivité ». The term aggressiveness used in the text aimed for translating the intensity of blackflies bites i.e. the biting rate
In the paper : we corrected as follow: “In this hotbed, a study showed low transmission of Onchocerciasis and a high biting rate in 2012 i.e. the biting rate were about 12 times higher than that observed 27 years ago, in the same focus”.
Line 117 and line 122 It is not APOC activities but CDTI
Answer: We deleted APOC and putted CDTI as suggested.
Line 134. What do you mean by theoretical saturation?
Answer: Theoretical saturation is an assumption made by researchers using research methods that after a number of focus groups or in-depth interviews, data collection no longer produces any new information. At this point, participants start repeating themselves or saying almost the same thing so that there is then redundancy in the data. This situation leads to the discontinuance of data collection.
Line 170: Explain your data analysis in more detail.
Answer: What is reported in the paper is detail for data analysis. All steps are mentioned (transcription, familiarization, constitution of coding sheet, coding and organization of data using a software, synthesis, writing report).
Could you please give more detail about your question on this point ?
Line 198: It would be good to know what proportion of the population perceive that onchocerciasis is related to sorcery and whether levels of economic development or education have any impact of perceptions. The Binza Ozone region also has some affluent areas.
Answer: The sampling strategy used in this study was a purposive sampling strategy which is not a random sampling strategy and the sample obtained is not representative of the population. Therefore, it is difficult to provide such a proportion. The phenomenon concerns mostly people living along Congo river and in all sectors of Binza Ozone. Recent studies showed an expansion of population as new houses are built and more and more people take house in this municipality. Currently, the township is divided newly with a real estate boom (boom immobilier)
Line 423 424 You have used the term Community Distributors CDs which is fine. Correct CDs in the following line
Answer: We corrected the sentence putting CDs as suggested.
Reviewer 2 Report
REVIEW FOR THE MANUSCRIPT : Community Participation in and Perception of Community-Directed Treatment with Ivermectin in Kinshasa, DRC
Manuscript ID : tropicalmed-526196
General comments
This is a pertinent and useful research documenting the reasons for low CDTI uptake in Kinshasa using qualitative methods. The findings could inform stakeholders and policy makers on how better engage the community to ensure optimal ivermectin use and consequently, fight against onchocerciasis. Overall, the methods are adequate, the findings are relevant and the conclusions are backed up by the study results.
As a general suggestion: Please thoroughly review the English language in the manuscript.
ABSTRACT
Line 24 sounds awkward (and well as…). Please rephrase.
INTRODUCTION
Line 41-43 : The information provided may not be very accurate. Kindly check the updated statistics from WHO : https://www.who.int/news-room/fact-sheets/detail/onchocerciasis
Line 55 : change ‘amount’ to ‘number’
METHODS
Line 101 : The expression « high rate of aggression » is unclear. Are we talking of blackflies ? Please rephrase accordingly.
Line 107 : Please correct community ‘relays’ (plural) to ‘relay’ (singular)
RESULTS
Line 224 : Does this expression « the stains left on the body following wounds due to scratching » refer to scratch marks ? Kindly clarify or rephrase sentence.
Line 245 : please replace ‘stings’ by ‘bites’
Line 260 : number of cases in the environment or in the community ?
Line 278 : Sentence is unclear, please rephrase.
DISCUSSION
Line 384 : Please correct Associated to « associate »
Line 397-397 : Are there studies which investigated the cause of these very frequent side effects ? High baseline microfilarial densities ? Co-infection with Loa loa ? A scientific explanation of the cause could help provide a solution and consequently increase CDTI uptake. It would be interesting to discuss in this light.
Line 420-421 : Indeed, health education is essential to ensure satisfactory adherence to CDTI by the population. A similar observation was made by Mukendi et al (Int J Infect Dis 2019; 79: 187–94. DOI: 10.1016/j.ijid.2018.10.021), who found that people with epilepsy in Aketi (DRC) are very willing to take ivermectin because they believe it alleviates their seizures, contrary to observations from Cameroon (Siewe et al, Epilepsy Behav 2019; 90: 70–8), where CDTI coverage among persons with epilepsy is very low because they are educated that epilepsy is a contraindication to ivermectin.
Another important point to be discussed is the fact that young people don’t know that onchocerciasis can cause blindness, but the old people have witnessed it in the past. This is because prior to CDTI, infection with O. volvulus was intense and chronic and usually progressed to blindness. However, it has been shown that ivermectin use has drastically reduced the disease burden of onchocerciasis by averting hundreds of thousands of potential cases of blindness in treated populations (https://www.who.int/news-room/fact-sheets/detail/onchocerciasis). Therefore, those born recently cannot be aware of how frequent blindness was in the past.
Author Response
Author's Reply to the Review Report (Reviewer 2)
General comments
This is a pertinent and useful research documenting the reasons for low CDTI uptake in Kinshasa using qualitative methods. The findings could inform stakeholders and policy makers on how better engage the community to ensure optimal ivermectin use and consequently, fight against onchocerciasis. Overall, the methods are adequate, the findings are relevant and the conclusions are backed up by the study results.
As a general suggestion: Please thoroughly review the English language in the manuscript.
Answer: we’ve reviewed English language in the manuscript.
ABSTRACT
Line 24 sounds awkward (and well as…). Please rephrase.
Answer: we rephrased as follows: “CDTI is considered devastating because of adverse effects of Ivermectin as well as inefficient after occurrence of deaths. This may explain the low level of community adhesion to this strategy”.
INTRODUCTION
Line 41-43: The information provided may not be very accurate. Kindly check the updated statistics from WHO: https://www.who.int/news-room/fact-sheets/detail/onchocerciasis
Answer: we’ve checked the updated statistics from WHO and have corrected this part as follows: “Onchocerciasis is one of eleven neglected tropical diseases (NTDs), recently targeted by the World Health Organization (WHO) for elimination out of the twenty existing ones, affecting poor populations living in the tropics and subtropics areas [6]. According to the WHO, around 120 million people worldwide are at risk of Onchocerciasis, among which 96% are in Africa [7]. Out of the 38 countries with endemic Onchocerciasis, 31 are located in sub-Saharan Africa, 6 in North and South America and 1 in Asia [8]. More than 99% of infected people live in 31 African countries. The Global Burden of Disease Study estimated in 2017 that there were 20.9 million infections O. volvulus prevalent worldwide: 14.6 million of the infected people had skin disease and 1.15 million had vision loss [8]”
Line 55: change ‘amount’ to ‘number’
Answer: we’ve changed “amount” to “number” as suggested
METHODS
Line 101: The expression « high rate of aggression » is unclear. Are we talking of blackflies? Please rephrase accordingly.
Answer: You’re right. The « high rate of aggression » is not the correct translation of French terms « taux d’agressivité ». The term used to translate the intensity of blackflies bites i.e. the biting rate
So, we’ve corrected as follows: “In this hotbed, a study showed low transmission of Onchocerciasis and a high biting rate in 2012 i.e. the biting rate were about 12 times higher than observed 27 years ago, in the same focus”.
Line 107: Please correct community ‘relays’ (plural) to ‘relay’ (singular)
Answer: we’ve corrected with the right translation of “relais communautaire”. So we putted “community volunteer”.
RESULTS
Line 224: Does this expression « the stains left on the body following wounds due to scratching » refer to scratch marks? Kindly clarify or rephrase sentence.
Answer: You’re right. That expression refers to scratch marks and we corrected the sentence as follows: “young girls are not only concerned with the nuisance caused by the bites of “Mbitiri” but also the stains left on the body following wounds due to scratch marks”.
Line 245: please replace ‘stings’ by ‘bites’
Answer: we replaced “stings” by “bites”.
Line 260: number of cases in the environment or in the community?
Answer: we replaced “environment” by “community”
Line 278: Sentence is unclear, please rephrase.
Answer: we rephrased the sentence as follows: “Coordination and Services of NOCP’s CDTI strategy to eradicate Onchocerciasis has set up an annual treatment in each zone where Onchocerciasis has been declared endemic. However, there is not a great uptake in Binza Ozone HZ to motivate community members to take Ivermectin”.
DISCUSSION
Line 384: Please correct Associated to « associate »
Answer: we replaced “associated” by “associate” as suggested.
Line 397-397: Are there studies which investigated the cause of these very frequent side effects? High baseline microfilarial densities? Co-infection with Loa loa? A scientific explanation of the cause could help provide a solution and consequently increase CDTI uptake. It would be interesting to discuss in this light.
Answer: the causes of serious adverse reactions related to ivermectin are poorly documented in the literature. Only Co endemicity onchocerciasis loiasis is more frequently reported as a cause of serious adverse events after taking ivermectin. The ideal solution to avoid its adverse effects and increase population adhesion would be to first identify the coendemic area and adapt treatment.
We discussed this part as follows: “An experimental study showed that the dominant clinical symptoms of adverse effects and toxicity of ivermectin in animals which often lead to mortality are tremors, ataxia, central nervous system depression and coma. This study does not show the cause of its side effects [30]. Loiasis is a major obstacle to ivermectin treatment for onchocerciasis control and lymphatic filariasis elimination in central Africa. In communities with a high level of loiasis endemicity, there is a significant risk of severe adverse reactions to ivermectin treatment. Information on the geographic distribution of loiasis in Africa is urgently needed but available information is limited. [31]. The real causes of these serious adverse reactions are poorly documented in the literature. However, according to existing studies, co-endemicity onchocerciasis loiasis is the main and most frequent cause of serious adverse events following ivermectin administration. Kuesel confirmed that the severe adverse reactions to ivermectin in people highly infected with Loa loa also delay progress towards elimination of lymphatic filariasis (LF) in Africa where the ’standard of control’ is co-administration of ivermectin and albendazole. People with LF do not directly benefit from ivermectin treatment since the LF’s symptoms are due to the macrofilariae. Consequently, use of ivermectin in areas co-endemic to LF and Loa loa, but not for least mesoendemic for onchocerciasis, does not have an acceptable risk-benefit ratio. The provisional strategy for LF control in Loa loa co-endemic areas is twice yearly treatment with 400 mg albendazole complemented with vector control (World Health Organization, 2012b). This strategy is now supported by the results of a community trial of semiannual MDA with 400 mg albendazole. The study showed that the middle rate geometric microfilariae level 12 months after the first treatment was reduced by 60% relative to the pre-treatment level. Co-infections with other filariae do not appear to significantly affect the adverse reaction profile of ivermectin, probably due to the fact that parasite numbers are lower than those in heavy Loa loa infections[32]”.
Line 420-421: Indeed, health education is essential to ensure satisfactory adherence to CDTI by the population. A similar observation was made by Mukendi et al (Int J Infect Dis 2019; 79: 187–94. DOI: 10.1016/j.ijid.2018.10.021), who found that people with epilepsy in Aketi (DRC) are very willing to take ivermectin because they believe it alleviates their seizures, contrary to observations from Cameroon (Siewe et al, Epilepsy Behav 2019; 90: 70–8), where CDTI coverage among persons with epilepsy is very low because they are educated that epilepsy is a contraindication to ivermectin.
Answer: Many thanks, we added this part in the text as suggested.
Another important point to be discussed is the fact that young people don’t know that onchocerciasis can cause blindness, but the old people have witnessed it in the past. This is because prior to CDTI, infection with O. volvulus was intense and chronic and usually progressed to blindness. However, it has been shown that ivermectin use has drastically reduced the disease burden of onchocerciasis by averting hundreds of thousands of potential cases of blindness in treated populations (https://www.who.int/news-room/fact-sheets/detail/onchocerciasis). Therefore, those born recently cannot be aware of how frequent blindness was in the past.
Answer: Many thanks, we discussed about it as follows: “Due to the lack of awareness campaigns and even less health education, young people don’t know that onchocerciasis can cause blindness, whereas old people have witnessed it in the past. Indeed prior to CDTI, infection with O. volvulus was intense and chronic and usually progressed to blindness. However, it has been shown that use of ivermectin has drastically reduced the disease burden of onchocerciasis by averting hundreds of thousands of potential cases of blindness in treated populations As a result, young people and recently born people cannot know how frequent blindness was in the past. Hence awareness and education campaigns are essential [8]”.